# Latency NMS Attacks:
# Is It Real Life or Is It Just Fantasy?

**Jean-Philippe Monteuuis**      **Cong Chen**      **Jonathan Petit**
Qualcomm Technologies, Inc.
{jmonteuu, congchen, petit}@qti.qualcomm.com

## Abstract

"Caught in a landslide, no escape from reality" summarizes the state of the research in AI offense: an attack might work on paper but does not necessarily in practice. In the last 5 years, we have seen the rise of latency attacks against computer vision systems. Most of them targeted 2D object detection, especially its Non-Max-Suppression (NMS) block, via adversarial images. However, we uncovered that, when tested in realistic deployment settings, the NMS latency attacks, accepted to top conferences, have very limited negative effects. In this paper, we define an evaluation framework (EVADE) to assess the practicality of attacks, and apply it to state-of-the-art NMS latency attacks. Attacks were tested on different hardware platforms, and different model formats and quantization. Results show that these attacks are not able to generate the claimed latency increase, nor transfer to other models (from the same family or not). Moreover, the latency increases remain within the latency requirements of downstream tasks in our evaluation, suggesting limited practical impact under these conditions. We also tested three defenses, which were successful in mitigating the NMS latency attacks. Therefore, in their current form, NMS latency attacks are just fantasy.

## 1   Introduction

Perception systems use camera inputs to perform tasks such as object detection, object classification, or segmentation. They rely on the performance of machine learning models to offer proper services. Especially, the latency of perception output plays an important role in delivering satisfactory user experience or safety-critical functions.

The maximum latency for real-time object detection models depends on the target application. For example, for autonomous driving, the maximum latency is typically around 100 ms to ensure safety and responsiveness (Lin et al., 2018).For surveillance systems, a latency of up to 200 ms is acceptable, as these systems often prioritize accuracy over immediate response (Hussain, 2024). In industrial robotics, latencies of up to 100 ms are often acceptable, but for more precise tasks, lower latencies around 10-20 ms are preferred (Vijayakumar and Vairavasundaram, 2024).

Most of the object detection models use Non-Max Suppression (NMS) to process all the candidate bounding boxes and output the most probable ones. It has been demonstrated that an attacker could craft perturbations to create many high-probability bounding boxes that would not be filtered by the NMS, and hence, increase the latency Wang et al. (2021).

However, as we are going to demonstrate in this paper, the prior art omitted many important aspects of deployed ML models that seriously question the relevance of current NMS latency attacks. More precisely, we tackle the following research questions.

- **RQ1**: How does the **hardware** affect the Attack Success Rate (ASR) of latency attacks against NMS?

39th Conference on Neural Information Processing Systems (NeurIPS 2025).

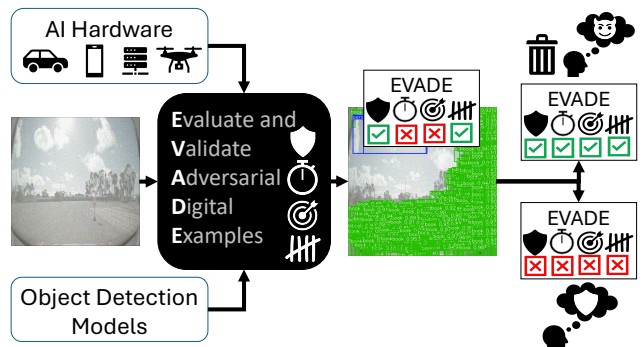

Figure 1: Overview of our evaluation framework (EVADE) to assess practicality of attacks. An input image is adversarially perturbed (to aim at increasing latency here). EVADE tests different defenses, model formats, checks impact on performance (e.g., accuracy, number of bounding boxes) to assess the attack risk level.

- **RQ2**: How does neural network **architecture** affect the ASR of NMS latency attacks?
- **RQ3**: How does the **size** of the model affect the ASR of NMS latency attacks?
- **RQ4**: When deployed, the models are compiled to a target format (e.g., QNN, ONNX, TensorRT). Does the **model format** affect the ASR of NMS latency attacks?
- **RQ5**: Models deployed on device are quantized to optimize their performance (e.g., memory and latency). How does **quantization** affect the effectiveness of NMS latency attacks?
- **RQ6**: How does "**Domain Shift**" affect the ASR of universal NMS latency attacks?
- **RQ7**: How applicable are existing **defenses** (e.g., adversarial training, compression, purification) against NMS latency attacks?

To answer these research questions and assess the practicality of NMS latency attacks, we designed a framework to **E**valuate and **V**alidate **A**dversarial **D**igital **E**xamples (EVADE)–see Figure 1. Then, we devised experiments (see Section 4) that demonstrated the lack of effectiveness, transferability, and robustness of state-of-the-art NMS latency attacks. Especially, NMS latency attacks did not succeed when NMS runs on GPU (which is commonly the case), and NMS latency attacks were not transferable across different models (size, or version from the same family), nor able to bypass existing defenses.

Our contributions are as follows:

- The EVADE framework that supports evaluation of practicality of attacks. We applied EVADE to latency NMS attacks.
- Generation of 114 adversarial datasets to support reproducibility and comparison of future research.
- Comprehensive evaluation of 7 hardware platforms, 4 latency attacks on 15 models, and against 3 defenses.

## 2 Related Work

### 2.1 Latency Attacks

There have been several papers on the topic of NMS latency attacks, which aim to increase the processing time of the Non-Max-Suppression (NMS) [1]algorithm by fulfilling two objectives. First, maximize the number of bounding box proposals being processed by NMS, and second, minimize the number of proposals being filtered during NMS. As a result, the high number of box proposals increases the NMS processing time, and thus, results in a delay of the Object Detection pipeline.

---

[1]More details on NMS can be found in Appendix A.

Table 1: Our evaluation of NMS Attacks compared to SOTA

| Evaluation Criteria | D | O | PS | BPS | **EVADE** |
|---|---|---|---|---|---|
| Transferability of the attack across different Hardware | ✗ | 4 | 2 | ✗ | **7** |
| Transferability of universal patch attack across datasets (Domain shift) | NA | NA | ✗ | ✗ | ✓ |
| Transferability across different export formats of a model | ✗ | ✗ | ✗ | ✗ | ✓ |
| Transferability across different models (sizes and architectures) | 4 | ✗ | 3 | ✗ | **10** |
| Transferability from a non-quantized model to quantized models | ✗ | ✗ | ✗ | ✗ | ✓ |
| Robustness against defenses | ✗ | ✗ | ✗ | ✗ | ✓ |

Looking at the prior art [2], there are two categories of NMS latency attack: *unique patch* and *universal patch*.

*Unique patch attacks* aim to craft a perturbation patch to be added "on the top of" a genuine image by optimizing, through iteration, the perturbation for this single genuine image. *Daedalus* (D) Wang et al. (2021) and *Overload* (O) Chen et al. (2024) are state-of-the-art unique patch NMS latency attacks.

*Universal patch attacks* aim to craft a perturbation patch to be added "on the top of" a genuine image by optimizing, iteratively, the perturbation across multiples images (dataset). From an attacker perspective, the advantage of *universal patch attacks* is to use a single perturbation that works for multiple images. *Phantom Sponges* (PS) Shapira et al. (2023) and *Beyond Phantom Sponges* (BPS) Schoof et al. (2024) are state-of-the-art universal patch NMS latency attacks.

## 2.2 Evaluation of Adversarial Attacks

The evaluation of digital adversarial attacks is a well-explored topic. For instance, several works have been released such as Adversarial Robustness Toolbox (ART) Nicolae et al. (2018) or RobustBench Croce et al. (2020). However, these works do not evaluate adversarial examples from a feasibility perspective. Indeed, current evaluation frameworks are limited. For instance, it is well-known that an attack optimized on a non-quantized model may not transfer to its quantized counterpart Li et al. (2024); Shrestha and Großmann (2024). Such an aspect is mandatory to assess the feasibility of the attack because models running on device are quantized. Therefore, it is important to know if the attacker can use a surrogate non-quantized model to perform its attack or if the attacker needs to optimize the attack on the quantized model. A follow-up feasibility aspect to consider is the transferability of the attack on the same model but with different sizes. Concretely, it is common practice to train model with different sizes (e.g., small, medium, large) Jocher et al. (2023). Therefore, one may wonder if attacks transfer successfully to any model from the same family. In Table 1, we summarize the evaluation criteria used in the four NMS latency attack papers. In comparison, our framework (EVADE) offers a comprehensive evaluation of the practicality of the attacks.

## 3 Evaluation Framework: EVADE

To assess the feasibility and practicality of NMS latency attacks, we designed an evaluation methodology as shown in Figure 2. First, we describe the components that will be tested (hardware, model family, model size, model format, model quantization, domain shift, defenses). Then, we describe the datasets, attacks, and defenses specifically used for our NMS latency attack testing. Finally, we explain the relevant metrics.

### 3.1 Components

**Hardware:** The *Hardware* component aims to solely evaluate the processing time of the NMS component. Since NMS is the component targeted by the attack, its processing time is supposed to increase greatly under attack compared to the genuine scenario (no attack). Knowing that the processing time of NMS is hardware dependent, our evaluation considers a wide range of hardware to cover all kind of use cases such as edge device, desktop, and cloud server. For instance, the attack may not affect a model running on a cloud server because of its powerful hardware. However, an

---

[2]For the sake of reproducible and fair evaluation caused by re-implementation errors, we discard work without publicly available code. However, those work are cited in Appendix C.

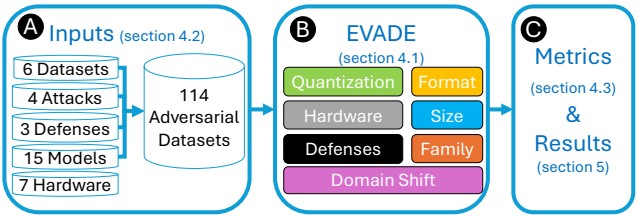

Figure 2: Proposed evaluation framework (EVADE) and corresponding sections where each phase is discussed. Block A shows the inputs used to perform the component-level assessment. Numbers represent what was used to assess NMS latency attacks. Block B lists the components evaluated.

edge device may be impacted by the attack because it may not have a powerful hardware to run the model.

**Model Family:** This component aims to evaluate if an attack optimized for a specific version of the model family (e.g., YOLOv8) can affect a different version of the model (e.g., YOLOv11). If an attack, optimized for a given model version, works for all versions of YOLO, then the attacker does not need to generate attack on each model version.

**Model Size:** This component aims to evaluate if an attack optimized for a specific size of the model (e.g., nano) can affect a model with a different size (e.g., large). If an attack, optimized for a given model size, works for any size of the targeted model, then the attacker does not need to generate attack on each model size.

**Model Format:** This component aims to understand if an attack optimized for a model under a specific framework (e.g., PyTorch) can affect a model exported to different framework (e.g., CoreML). For instance, it is common to have the model specification, training, and evaluation being done under PyTorch or TensorFlow. However, to deploy models on iPhone, they must be exported to a framework different than PyTorch, namely CoreML. If an attack, optimized for a model under one framework, works for any framework, then the attacker does not need to worry about model export.

**Model Quantization:** This component aims to understand if an attack optimized for a non-quantized model works against a quantized version of this model. The motivation behind this evaluation is that all latency attacks have been developed for non-quantized model. However, models are quantized to reduce their size and improve their efficiency, particularly for deployment on resource-constrained devices like mobile phones or edge devices. Moreover, quantization has demonstrated some negative and positive effect on attack success rate Bernhard et al. (2019). Thus, one must test attacks against quantized models to assess its practicality.

**Domain Shift:** This component aims to assess the universality of universal latency attacks. As a reminder, universal attacks aim to create a unique patch that fools any images. For instance, an attacker generates a universal patch optimized on an autonomous driving dataset and on an object detection model. We test this patch by applying it on "unseen" images and check if the attack still works.

**Defenses:** This component aims to understand if latency attacks are robust against existing defenses. We recommend to focus on diversifying the categories of defenses for the evaluation instead of having multiple defenses from the same category. Indeed, there is little to no interest to evaluate two defenses that are fundamentally similar because if the attacks fails against one defense, then the attack will fail against the second defense.

## 3.2   Inputs

To evaluate the feasibility of NMS attacks, EVADE requires a diversity of inputs such as genuine and adversarial datasets, models, NMS attacks, different hardware, and defenses. The following sections describe each input.

**Datasets:** For all our experiments, we use COCO 2017 validation as our baseline dataset. COCO is the standard large-scale dataset for object detection with 5K images. One main reason for choosing COCO 2017 validation as our baseline dataset is the heterogeneity of the images. In other words, the images' semantics are diverse and provide more entropy.

**Adversarial Datasets:** As done in the prior art Chen et al. (2024); Wang et al. (2021); Shumailov et al. (2021); Schoof et al. (2024), the common practice to evaluate a latency attack is to generate an adversarial dataset. The process of creating an adversarial dataset is to generate a unique perturbation (optimized for a target AI model) for each image, repeated N times (where N is the requested number of iterations for optimizing the perturbation). In a nutshell, each adversarial dataset contains adversarial images optimized using a given latency NMS attack and a given targeted model. In this paper, we generated 114 adversarial datasets to test attack effectiveness and transferability.

**Models:** To match the experiments from the state-of-the-art on NMS latency attacks, we use the same family of model (YOLO). We test different versions (v3, v5, v8, v11, v12), sizes (nano, small, medium, large, extra-large), export formats, and quantization schemes.

**Attacks:** Our focus is to evaluate latency NMS attacks. Therefore, in this paper, we evaluate the four state-of-the-art NMS latency attacks, namely, Daedalus (D), Overload (OL), Phantom Sponge (PS), and Beyond Phantom Sponge (BPS). We reused the settings defined in their respective paper[3].

**Defenses:** To assess existing defenses against NMS latency attacks, we selected three categories of defenses: an active defense, a passive defense, and a natural defense. An active defense is an extra component added to the inference pipeline (e.g., a dedicated defensive AI model). A passive defense is a defense that does not tamper with the image being processed during inference. An example of passive defense can be a filtering technique such as limit the number of bounding boxes being processed by NMS Chen et al. (2024). Lastly, a natural defense uses data processing schemes within the inference pipeline that may affect the attack. For instance, data transformations such as image compression or image resizing would fall under natural defenses.

## 3.3 Metrics

**Number of bounding boxes**: When it comes to evaluating NMS latency attacks, the evident metric is the *inference time*. However, the *inference time* is influenced not only by the attack, but also by the hardware used to run the model (CPU and GPU), the type of model being used for the experiment, and quantization. Therefore, to independently evaluate each component discussed in Section 3.1 (except hardware), we set the number of bounding boxes after NMS as our core metric. Indeed, we found out that the number of bounding boxes was reproducible when testing the same model on the same dataset but with different hardware. Also, the *number of bounding boxes* is highly correlated to the outcome of the attack.

**Visualization of attack effectiveness**: We wanted to provide a visual interpretation of the effect of the attack. For each table, we follow a four color scale: green, yellow, orange, and red. Green and red highlight the extreme cases such as when the attack failed or is able to attain the maximum NMS detection. Yellow and orange reflect intermediate cases such as an abnormal number of bounding boxes but the attack is not at its full potential. For most of the evaluations, we divided the maximum number of bounding boxes allowed by the model (300 boxes) into 4 ranges such as green (0-75), yellow (76-125), orange (126-225), red (226-300).

# 4 Evaluation

In this section, for each component, we first detail the experiment, and then discuss the results. We remind the reader that the objective of the evaluation is to assess the attack practicality. Understanding the reason behind the results is out of scope. In the following Tables, we abbreviated the attack names when needed and as follows: G (Genuine), D (Daedalus), O (Overload), PS (Phantom Sponge), BPS (Beyond Phantom Sponge). Each cell with a numerical value represents the round up average number of bounding boxes for a targeted dataset.

## 4.1 Hardware (RQ1)

---

[3]Details about each attack and their parameters can be found in Appendix C.

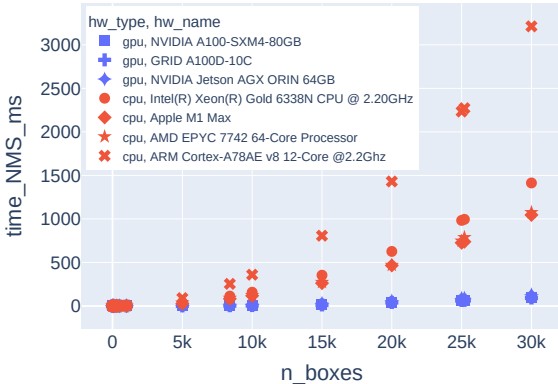

Figure 3: Inference time of NMS (ms) for different hardware platform in function of the number of bounding boxes that went through the NMS process. Each data point uses 30k bounding boxes proposals before NMS.

Table 2: Theoretical evaluation of attack success if NMS outputs 25k (yolov5) or 8k (yolov8) bounding box proposals.

| Hardware | | | Attack Success (ms) | | | |
|---|---|---|---|---|---|---|
| type | name | use case | 100 | 500 | 1000 | 2000 |
| GPU | NVIDIA A100-SXM4-80 GB | Server | ✓ ✓ | | | |
| | GRID A100D-10C | Desktop | ✓ ✓ | | | |
| | NVIDIA Jetson AGX ORIN 64GB | Edge-device | ✓ ✓ | | | |
| CPU | Intel Xeon Gold 6338N | Desktop | ✓ ✓ | ✓ | ✓ | |
| | Apple M1 Max | Laptop | ✓ ✓ | ✓ | | |
| | AMD EPYC 7742 | Server | ✓ ✓ | ✓ | | |
| | ARM Cortex-A78AE | Edge-device | ✓ ✓ | ✓ | ✓ | ✓ |

**Experiment**: Our focus is to understand how the number of bounding box affects the inference time of NMS. To this end, we use the implementation of NMS from Ultralytics Jocher et al. (2023), which is derived from the official PyTorch NMS implementation Paszke et al. (2019). [4]

**Results**: Figure 3 shows the NMS inference time in function of the number of bounding box proposals. We plot the results for the seven hardware configurations. We observe that the NMS attack is the most successful when the NMS computation is performed on CPU with a NMS inference time reaching more than 3 seconds in one instance. Now, if we want to translate Figure 3 into an analysis of the feasibility of the attack, then Table 2 would be the closest answer.

> **Answer to RQ1**: NMS latency attacks cannot achieve the latency increase needed to affect applications when the NMS processing runs on GPU. The attack must generate more than 10k bounding box proposals to start seeing a latency greater than 500 ms on CPU. However, models can cap the maximum number of proposals to a lower number, making the attack unsuccessful.

### 4.2 Model Family Transferability (RQ2)

**Experiment**: We use COCO 2017 validation dataset, 4 attacks, and 5 model versions of YOLO (v3, v5n, v8n, v11n, v12n). We generate 20 adversarial datasets (4 attacks × 5 models). In total, we run 100 experiments (20 adversarial datasets × 5 models).

**Results**: Table 3 shows the effect of each latency attack on different model version from the same family (YOLO). Results show that the attacks do not transfer across model versions of YOLO. Each attack only works when the source model and the target model are the same. From an attacker perspective, it means she must craft adversarial examples for each model version.

---

[4]More details of bounding box generation algorithm can be found in Appendix D.

Table 3: Evaluation of each attack's transferability across the YOLO model family. The adversarial dataset is a perturbed ***COCO 2017 validation***. Each cell with a numerical value represents the round up average number of bounding boxes for a targeted dataset. Colors include green (0-75), yellow (76-125), orange (126-225), red (226-300). **G** stands for Genuine. **TLDR: NMS attacks do not transfer across different version of the same model**.

| Targeted Model Version | G | Datasets | | | | | | | | | | | | | | | | | | | |
|---|---|---|---|---|---|---|---|---|---|---|---|---|---|---|---|---|---|---|---|---|---|
| | | Daedalus | | | | | Overload | | | | | Phantom Sponge | | | | | Beyond Phantom Sponge | | | | |
| | | v3n | v5n | v8n | v11n | v12n | v3n | v5n | v8n | v11n | v12n | v3n | v5n | v8n | v11n | v12n | v3n | v5n | v8n | v11n | v12n |
| v3n | 7 | 257 | 9 | 8 | 7 | 8 | 300 | 7 | 8 | 7 | 7 | 212 | 6 | 7 | 7 | 7 | 116 | 7 | 7 | 7 | 7 |
| v5n | 6 | 6 | 300 | 7 | 6 | 7 | 5 | 300 | 6 | 5 | 5 | 5 | 273 | 4 | 5 | 4 | 6 | 186 | 5 | 5 | 4 |
| v8n | 6 | 7 | 14 | 247 | 6 | 9 | 5 | 7 | 300 | 5 | 6 | 6 | 4 | 228 | 4 | 4 | 6 | 4 | 151 | 5 | 4 |
| v11n | 7 | 7 | 10 | 7 | 235 | 9 | 6 | 7 | 8 | 300 | 7 | 6 | 5 | 4 | 158 | 4 | 7 | 5 | 6 | 124 | 4 |
| v12n | 6 | 6 | 7 | 6 | 6 | 284 | 5 | 4 | 4 | 6 | 300 | 5 | 4 | 4 | 5 | 195 | 6 | 4 | 5 | 5 | 218 |

Table 4: Evaluation of each attack's transferability across different model (YOLOv8) sizes. **TLDR: NMS attacks do not transfer across different size of the same model**.

| Targeted Model Size | G | Datasets (*COCO 2017 validation*) | | | | | | | | | | | | | | | | | | | |
|---|---|---|---|---|---|---|---|---|---|---|---|---|---|---|---|---|---|---|---|---|---|
| | | Daedalus | | | | | Overload | | | | | Phantom Sponge | | | | | Beyond Phantom Sponge | | | | |
| | | n | s | m | l | x | n | s | m | l | x | n | s | m | l | x | n | s | m | l | x |
| n | 6 | 233 | 7 | 7 | 7 | 6 | 300 | 5 | 5 | 6 | 5 | 224 | 5 | 5 | 6 | 6 | 152 | 5 | 6 | 6 | 6 |
| s | 4 | 8 | 214 | 8 | 8 | 8 | 9 | 300 | 8 | 7 | 7 | 6 | 192 | 7 | 7 | 7 | 7 | 183 | 7 | 7 | 7 |
| m | 6 | 8 | 12 | 206 | 9 | 12 | 9 | 18 | 300 | 16 | 11 | 7 | 8 | 222 | 8 | 8 | 8 | 9 | 192 | 8 | 8 |
| l | 7 | 8 | 11 | 49 | 249 | 70 | 9 | 13 | 73 | 300 | 44 | 7 | 8 | 9 | 219 | 8 | 8 | 8 | 12 | 172 | 8 |
| x | 5 | 8 | 10 | 72 | 112 | 276 | 9 | 14 | 103 | 126 | 300 | 8 | 9 | 11 | 9 | 212 | 8 | 8 | 14 | 10 | 116 |

> **Answer to RQ2**: NMS latency attacks do not transfer between YOLO model versions. The attacker must know the target model. The question remains if this result applies to another model families.

## 4.3 Model Size Transferability (RQ3)

**Experiment**: We follow the same experiment as in Section 4.2 but with five sizes of YOLOv8 (nano, small, medium, large, XL).

**Results**: Table 4 shows that latency attacks do not transfer across different size of the model. Each attack only works when the source model and the target model are the same. We observe mild transferability of Daedalus and Overload from YOLOv8x to YOLOv8l, but insufficient to reach the critical latency threshold (i.e., cells are orange or yellow). This means the attacker must know the target model size.

> **Answer to RQ3**: NMS latency attacks do not transfer to other sizes of the same YOLO model.

## 4.4 Model Export Transferability (RQ4)

**Experiment**: We use 5 datasets (one genuine and four adversarial), 3 export formats of the model YOLOv8n (onnx, coreml, and openvino). *CoreML* is the model format used for running AI model on Apple's AI processor. *OpenVINO* is the model format optimized for Intel's processor. *ONNX* is an open-source format. We generate four adversarial datasets (4 attacks × 1 model) based on COCO 2017 validation. In total, we run an inference on the 4 adversarial datasets per model format.

**Results**: Table 6 shows that latency attacks transfer across export model formats. Each attack generated on a PyTorch model format works equally well on the same model exported to ONNX, CoreML, or OpenVINO format. From an attacker perspective, the attacker does not need to worry about the model format.

> **Answer to RQ4**: Model format does not affect the attack success. The model conversion (i.e., optimized operation, compilation) does not provide intrinsic robustness against NMS latency attacks.

## 4.5 Model Quantization Transferability (RQ5)

**Experiment**: We use 5 datasets (one genuine and four adversarial), 4 models based on YOLOv8n (non-quantized PyTorch model, non-quantized OpenVINO model, quantized GPU model (TensorRT, PTQ, INT8 activations), quantized CPU model (OpenVINO, PTQ, INT8). We generate four adversarial datasets (4 attacks × 1 model) based on COCO 2017 validation. In total, we run an inference on the 4 adversarial datasets per model.

Table 5: Evaluation of each attack's transferability when tested on quantized models. **TLDR: Overload is the only attack that perfectly transfers when targeting a quantized version of the source model**.

| Data type | Device type | G | D | O | PS | BPS |
|---|---|---|---|---|---|---|
| FP16 | CPU (OpenVino) | 6 | 233 | 300 | 224 | 152 |
| | GPU (TensorRT) | 6 | 233 | 300 | 224 | 152 |
| INT8 (PTQ) | CPU (OpenVino) | 6 | 230 | 300 | 216 | 143 |
| | GPU (TensorRT) | 4 | 195 | 300 | 89 | 67 |

Table 8: Domain Shift evaluation of universal patch attacks (*Phantom Sponge* and *Beyond Phantom Sponge*). **TLDR: Universal Patch attacks do not work under domain shift.**

| Dataset (Target) | | Genuine | Phantom Sponge | | | | | | Beyond Phantom Sponge | | | | | |
|---|---|---|---|---|---|---|---|---|---|---|---|---|---|---|
| | | | COCO | | BDD-100k | | nuImages | | COCO | | BDD-100k | | nuImages | |
| Name | Split | | val | train | val | train | val | train | val | train | val | train | val | train |
| COCO | val | 4 | 224 | 7 | 5 | 5 | 5 | 5 | 152 | 7 | 6 | 6 | 5 | 5 |
| | train | 6 | 6 | 193 | 3 | 13 | 3 | 3 | 4 | 163 | 3 | 8 | 3 | 3 |
| BDD-100k | val | 7 | 16 | 6 | 300 | 26 | 14 | 5 | 11 | 9 | 288 | 20 | 9 | 6 |
| | train | 6 | 10 | 5 | 36 | 300 | 6 | 4 | 9 | 7 | 9 | 296 | 6 | 5 |
| nuImages | val | 5 | 8 | 5 | 9 | 4 | 300 | 7 | 4 | 4 | 4 | 5 | 300 | 5 |
| | train | 5 | 4 | 2 | 4 | 3 | 9 | 300 | 4 | 4 | 4 | 3 | 7 | 300 |

**Results**: From Table 5, we can say that latency attacks transfer to quantized models[5]. Therefore, an attacker can create perturbations on non-quantized models and expect similar ASR on quantized models.

> **Answer to RQ5**: NMS latency attacks transfer from non-quantized to quantized models (INT8 Post-Training Quantization).

Table 6: Evaluation of each attack's transferability across model formats. **TLDR: NMS attacks transfer across different model formats**.

Table 7: Performances of each attack against three different types of defense. **TLDR: Each attack becomes harmless when facing a defense**.

| Role | Fomat Name | G | D | O | PS | BPS |
|---|---|---|---|---|---|---|
| Source | pytorch | 6 | 233 | 300 | 224 | 152 |
| Target | onnx | 6 | 233 | 300 | 224 | 152 |
| Target | coreml | 6 | 233 | 300 | 224 | 151 |
| Target | openvino | 6 | 233 | 300 | 224 | 152 |

| Defense | G | D | O | PS | BPS |
|---|---|---|---|---|---|
| None | 6 | 233 | 300 | 224 | 152 |
| PDM | 5 | 5 | 5 | 5 | 5 |
| JPEG | 6 | 8 | 55 | 5 | 6 |
| Maximum Detection (10) | 5 | 7 | 8 | 6 | 6 |

## 4.6 Domain Shift Transferability (RQ6)

**Experiment**: we use two universal patch attacks (Phantom Sponge and Beyond Phantom Sponge). For dataset, we use two automotive datasets BDD-100k and nuImages in addition to our core dataset COCO 2017. For each dataset, both training and validation splits are used in the experiment. As for adversarial datasets, we have 72 of them for this experiment. We generated 12 adversarial patches (2 attacks × 3 datasets × 2 splits) that are applied to a total of 6 genuine datasets (3 datasets × 2 splits).

**Results**: A seen in Table 8, both attacks do not transfer across dataset, and this, even within the same domain (e.g., BDD-100k to nuImages). Therefore, if an attacker wanted to create an adversarial patch on an automotive dataset and apply this patch on an image with an automotive context, then the attack will not work.

> **Answer to RQ6**: Universal patch latency attacks do not transfer to other datasets.

## 4.7 Defenses (RQ7)

**Experiment**: we use 5 datasets (one genuine and 4 adversarial), 1 model (YOLOv8n), and 3 defenses (PDM-Pure, JPEG, and maximum detection threshold). We set the maximum detection threshold to 10 based on the maximum number of objects present in individual images of COCO 2017 dataset. All 4 adversarial datasets (4 attacks × 1 model) are based on COCO 2017 validation. In total, we run 20 experiments (5 datasets × 3 defensive cases and one genuine case).

---

[5]Note that this paper did not aim at providing a comprehensive study of quantization techniques.

Table 9: EVADE: Overall evaluation results for latency NMS attacks. ✗/✓ means that the attack failed / succeeded when tested against this component. "N/A" means "Not Applicable".

| Attack | Hardware | Model | | | | Domain Shift | Defenses | Score | Result |
|--------|----------|--------|------|--------|--------------|--------------|----------|-------|--------|
| | | Family | Size | Format | Quantization | | | | |
| D | ✗ | ✗ | ✗ | ✓ | ✗ | N/A | ✗ | 1/6 | harmless |
| O | ✗ | ✗ | ✗ | ✓ | ✓ | N/A | ✗ | 2/6 | harmless |
| PS | ✗ | ✗ | ✗ | ✓ | ✗ | ✗ | ✗ | 1/7 | harmless |
| BPS | ✗ | ✗ | ✗ | ✓ | ✗ | ✗ | ✗ | 1/7 | harmless |

**Results**: Table 7 shows that all attacks were filtered by the three defenses. Since all attacks failed, there is no need to expand this evaluation with more defenses. A special mention goes to the attack *Overload* which shows some resistance against the JPEG defense.

> **Answer to RQ7**: NMS latency attacks couldn't defeat the defenses. New (realistic) attacks are needed.

### 4.8 Summary

As seen in Table 9, all attacks went through 6 evaluations (7 for universal attacks) and were found harmless in real-world scenario. The best attack is Overload, which passed only 2 out of 6 evaluations. However, this analysis demonstrates the lack of practicality of the current NMS latency attacks.

## 5 Discussion

### 5.1 Is NMS Latency Attack a Dead End?

Thanks to our evaluation framework EVADE, we uncovered many limitations of state-of-the-art latency NMS attacks. Actually, our results suggest that these attacks did not pose a practical threat when tested in realistic settings. That being said, because NMS attacks create unwanted bounding boxes, it would be interesting to analyze their impact on the processes after object detection such as Object Tracking.

Another idea for latency attacks is to target NMS-free object detection model (e.g., DETR and YOLOv10). Obviously, existing frameworks for generating latency NMS attack do not apply here because they require access to the number of proposals before NMS (which is not available in NMS-free models). One would have to devise a new attack and test it using EVADE.

### 5.2 Enhancing existing NMS

**Clarification on low transferability**  While low transferability may limit the generalizability of latency-based attacks, it does not imply that such attacks are ineffective. As demonstrated in Tables 3 and 4, the attack is successful when targeting the same model used to generate the adversarial input, consistent with white-box assumptions. Importantly, latency attacks remain a credible threat in scenarios where the target model is known or can be reasonably guessed.

**Boosting attack transferability**  Looking at the attacks' transferability in Sections 4.2 and 4.3, we can conclude existing NMS latency attacks do not transfer because they are white-box attacks. Therefore, the attacks work only against the targeted model. However, to fill this gap, an interesting future work could be to add techniques for transferability boosting (e.g., Quantization Aware Attack Yang et al. (2024) and Momentum-based iterative Dong et al. (2018)) in the attack generation pipeline of existing NMS latency attack and see if the transferability improves.

**Generating high-confidence fake boxes**  Most fake bounding boxes currently have lower confidence scores than genuine ones. As a result, defenses that limit the number of displayed boxes tend to retain genuine boxes, since they have higher confidence. If an attack could generate fake boxes with higher confidence than the genuine ones, the defense might mistakenly discard the genuine boxes.

**Study on quantization**    A comprehensive evaluation of diverse quantization techniques—including varying bit-widths, post-training quantization, quantization-aware training, and mixed-precision strategies—remains an important direction for future work to better understand their impact on NMS latency attack robustness.

### 5.3    Expand EVADE

We envision a couple of interesting additions to EVADE. First, we would like to add new metrics, such as image similarity score (e.g., SSIM, LPIPS) to evaluate how close are the adversarial images from the genuine ones. This would support assessing the perceptibility of the attack.

Second, EVADE could be applied to other vision models such as audio speech recognition or natural language processing. For instance, it has been shown that attacks targeting LLM are hardly transferable across models Lin et al. (2025); Liu et al. (2024) or do not work on a quantized version of the model Dong et al. (2025); Yi et al. (2025).

## 6    Conclusion

In recent academic papers, computer vision models have been shown to be vulnerable to latency attacks. Notably, object detection models were dramatically slowed down by adversarially-crafted perturbations that exploit the Non-Max Suppression algorithm. However, in this paper, we put the state-of-the-art NMS latency attacks to the test. We focused on investigating their practicality. We designed an evaluation framework (EVADE) to test key deployment components. We uncovered that NMS latency attacks suffer from a set of weaknesses, some specific to latency attacks, some valid for all digital adversarial examples. As demonstrated in this paper, without addressing the identified weaknesses, current NMS latency attacks targeting object detection have very limited practical impact, and hence, just fantasy.

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

# Appendix

# A    Non-Maximum Suppression (NMS)

Non-Maximum Suppression (NMS) is a post-processing step commonly used in object detection pipelines to prune redundant bounding boxes. During inference, a detection model often outputs multiple overlapping bounding boxes for the same object—each with its own location, size, and confidence score. NMS serves to suppress (i.e., remove) boxes whose overlap with a higher-confidence box is too large (above a predefined IoU threshold), thereby retaining only the most representative candidates.

## A.1    Background on NMS

Non-Maximum Suppression (NMS) is a post-processing step commonly used in object detection pipelines to prune redundant bounding boxes. During inference, a detection model often outputs multiple overlapping candidate bounding boxes for the same object–each with its own location, size, and confidence score. NMS takes all candidate bounding boxes and filters out the low-confidence ones, and boxes that have large IoU with higher-confidence boxes. NMS outputs the most representative candidate bounding boxes (also called *proposals*).

We would like to bring attention to two important parameters: the maximum number of bounding box proposals (set by the model), and the maximum detection (set by NMS). In the YOLO model family, the former parameter has been reduced with every version. In fact, YOLOv5 caps the number of proposals to 25,000, while YOLOv8 caps it to 8,000. PyTorch NMS implementation sets the maximum number of bounding box proposals to 30,000. As we will discuss in Section 4, this has a huge impact on the attack success. The maximum detection set by NMS defines the maximum number of proposals to output. This means that even if the attack forces the model to feed more bounding box proposals than what NMS accepts, it won't necessarily affect the latency.

In the standard implementation of NMS (Table 10), the maximum detection is set to 300. This parameter directly limits the effect of the fake bounding boxes on the downstream tasks (e.g., multi-object tracking Wang et al. (2024)). However, the four state-of-the-art NMS latency attacks (described in Section 2) consider a maximum detection of 30,000 which is not realistic in most use cases (e.g., autonomous driving, surveillance, or traffic monitoring).

## A.2    Parameters

In Table 10, we describe the parameters of NMS used in the hardware evaluation.

| Parameter | Description | Value |
|---|---|---|
| conf_thres | The confidence threshold below which boxes will be filtered out | 0.25 |
| iou_thres | The IoU threshold below which boxes will be filtered out during NMS | 0.45 |
| classes | A list of class indices to consider. If None, all classes will be considered | None |
| agnostic | If True, the model is agnostic to the number of classes. And all classes will be considered as one. | False |
| multi_label | If True, each box may have multiple labels. | False |
| labels | The list contains the apriori labels for a given image. | () |
| maximum detection | The maximum number of boxes to keep after NMS | 300 |
| nc | The number of classes output by the model. Any indices after this will be considered masks. | 0 |
| max_time_img | The maximum time (seconds) for processing one image. | 0.05 |
| max_nms | The maximum number of boxes into torchvision.ops.nms(). | 30000 |
| max_wh | The maximum box width and height in pixels. | 7680 |
| in_place | If True, the input prediction tensor will be modified in place. | True |
| rotated | If Oriented Bounding Boxes (OBB) are being passed for NMS. | False |
| end2end | If the model doesn't require NMS. | False |

Table 10: Parameters of Non-Maximum Suppression by Ultralytics

### A.3 NMS processing time

Figure 10 depicts the line of code (328) of the function named `non_max_suppression` located in `ultralytics.utils.ops.py` from the library Ultralytics containing the inference time for the NMS processing. For our hardware evaluation, we captured (`time.time()-t`) into a variable name `time_nms` which was returned by `non_max_suppression` in addition to the output of `non_max_suppression`. For those who wish to reproduce our approach, we use the version 8.3.34 of *Ultralytics*.

```
327             output[xi] = x[i]
328             if (time.time() - t) > time_limit:
329                 LOGGER.warning(f"WARNING ⚠ NMS time limit {time_limit:.3f}s exceeded")
330                 break  # time limit exceeded
331
332         return output
```

Figure 4: Location of inference time for NMS within the *Ultralytics* library

## B  Commands used for evaluation

All our commands are based on the documentation from the *Ultralytics* library.

### B.1  Format

to export our model to a different format, we use the mode *export* available with the model of our choice. Then, we specify the model format (e.g., ONNX) required for the export such as `model.export(format="onnx")`.

### B.2  Quantization

To quantize our model, we need to export a model in a specific format (like in the previous section). Then, we need to enable the int8 quantization during the model export. For instance, if we want a quantized int8 version of *OpenVINO*, then we just need to run the following command such as `model.export(format="openvino", int8=True, nms=True)`. The *nms* argument includes the NMS post-processing in addition to the quantized model during the export process.

## C  NMS Latency Attacks

### C.1  Prior Art

Previous work on latency attacks exploits the observation that generating numerous bounding boxes resistant to NMS elimination. There is an extensive literature on NMS Latency Attacks targeting Object Detection such as backdoor attack Xiao et al. (2024), bit-flip attack Sistla et al. (2025), and adversarial inputs attack. However, the first two categories rely on strong assumptions. For backdoor attacks, those assumptions include having access to the training dataset (which is not publicly disclosed) or being able to inject sufficient malicious data into the training dataset (which implies the attacker knows which data will be picked to train the model). For bit-flip attack, the attacker somehow has access to the weights of the model used in production and can tamper with the weights. In general, model weights are not easily accessible because they are private and there are plenty of security mechanisms to monitor any change made to the model such as cryptographic signature, encryption for data obfuscation, access control for people accessing the model weights. With such privileges, an attacker can have a better impact by using a malware attack or by leaking the model weights to the public or to competitors. Adversarial inputs attack, on the other hand, is based on more realistic assumptions such as tampering inputs at inference by probing a model in a product or by using a surrogate model that may be the same as the one in production.

For adversarial inputs attack, There are five attacks which focus on Camera Object Detection: Daedalus, Overload, Phantom Sponge, Beyond Phantom Sponge, and DetStorm.

Daedalus (D) Wang et al. (2021) pioneered latency attacks, using adversarial examples against NMS to achieve image-wide perturbation.

Overload (O) Chen et al. (2024), improved on DS by proposing a spatial attention mechanism, that prioritizes bounding box creation in less occupied areas to reduce overlap.

Phantom Sponge (PS) Shapira et al. (2023), advances latency attacks using universal adversarial perturbations (UAP) to apply a single pattern across multiple images, removing generation time.

Beyond Phantom Sponges (BPS) Schoof et al. (2024) is a spin-off of the original Phantom Sponge Attack. BPS replaces the bounding box area loss from PS with a new loss (named *IoU loss*). This new loss function decreases the IoU of a given bounding box and every other bounding boxes instead of just decreasing the bounding box area.

DetStorm (DS) Muller et al. (2025) is another spin-off of the original Phantom Sponge Attack. DS modifies the original PS attack to make it work as a physical adversarial example (real-world). Since DetStorm is based on PS, it has the same flaws as Phantom Sponge.

### C.2 Evaluation Settings

Each latency attack relies on a set of parameters and values that are common to all attacks but also specific to an attack (see Table 11). Here is a list of all the parameters with their description.

***maximum iterations*** is the maximum number of iterations allowed to optimize the adversarial noise to be added to the image.

***initial constant*** ($c_0$) is the value of the constant ($c$) at initialization when generating the Daedalus attack. The constant ($c$) helps to balance the distortion and the adversarial loss function related to the attack.

***binary search steps*** is the maximum number of search steps used during the binary search to find the best value for the constant $c$.

***confidence*** ($\gamma$) is the minimal threshold value for the confidence of adversarial bounding boxes.

***grid size*** is the size of the grids used to divide the image for the spatial attention mechanism Chen et al. (2024).

***epsilon*** is the maximum size of the perturbation.

***lambda 1*** ($\lambda1$) is a weighting factor affecting the *maximum object* loss and the *maximum IoU loss* Shapira et al. (2023); Schoof et al. (2024).

***lambda 2*** ($\lambda2$) is a weighting factor affecting the *bounding box area* loss Shapira et al. (2023); Schoof et al. (2024).

| Parameter | D | O | PS | BPS |
|---|---|---|---|---|
| max. iterations | 1000 | | | |
| initial constant ($c_0$) | 2 | NA | NA | NA |
| binary search steps | 5 | NA | NA | NA |
| confidence | 0.3 | NA | NA | NA |
| grid size | NA | 10x10 | NA | NA |
| $\epsilon$ | NA | 15 | 70 | |
| $\lambda1$ | NA | NA | 1 | |
| $\lambda2$ | NA | NA | 10 | |

Table 11: Attacks' parameters and values used in our paper. "N/A" means "Not Applicable".

## D Generating box proposals for NMS

For the hardware evaluation, we use Algorithm 1 to generate non-overlapping bounding boxes for a given image size ($W \times H$). The motivation to create such a script is because existing NMS attacks do not always For instance, if we use a $640 \times 640$ images, then we will generate 409600 boxes which is way above the maximum number of bounding box proposals that can be handled by Ultralytics NMS. Therefore, we make sure to cap our total number of proposals with the parameter $M$. By

default, $M$ is set to 30000 because it is the maximum number of proposals allowed to be processed by Ultralytics' NMS. Now, if we want to choose the number of proposals being processed by the NMS function, then we are using the parameter $N$. For instance, if the NMS must process 5000 proposals, then $N$ equals 5000. The remaining 25000 proposals will have their confidence value set to 0 and thus, they will be filtered out as part of the NMS pre-processing.

---

**Algorithm 1** Generate non-overlapping box proposals

---

**Require:** $W, H$: List
1: $\mathcal{B} \leftarrow \emptyset$            // Set of bounding boxes proposals
2: $M = 30000$            // Max. proposals for PyTorch's NMS
3: $w = 1$            // Width of the box
4: $h = 1$            // Height of the box
5: $n = 0$            // Number of proposals
6: $l = 1$            // Identifier of the object class for this proposal
7: **while** $n < M$ **do**
8:     **for** $h = 0 \rightarrow H$ **do**
9:         **for** $w = 0 \rightarrow W$ **do**
10:             $n = n + 1$
11:             $x = h + 0.5$            // Box's center (x) coordinate
12:             $y = w + 0.5$            // Box's center (y) coordinate
13:             **if** $n < N$ **then**
14:                     // Wanted number of proposals for NMS
15:                 $c = 1$
16:             **else**
17:                 $c = 0$
18:             **end if**
19:             $b = [x, y, w, h, c, l]$            // Define proposal box
20:             $\mathcal{B} \leftarrow b$            // Append the $b$ to $\mathcal{B}$
21:         **end for**
22:     **end for**
23: **end while**
24: **return** $\mathcal{B}$

---

# E Additional Details regarding the evaluation

## E.1 Hardware: measuring inference time

To measure the inference time (in milliseconds), we use the time computed in the Ultralytics NMS code (see Appendix A.3). Because existing attacks couldn't generate the maximum bounding box proposals of 30,000 and we need to evaluate the worst case scenario, we created a python script to generate a tensor of non-overlapping 30,000 bounding box proposals, where 30,000 is the maximum number of proposals allowed by the NMS function in the Ultralytics library. Then, for each proposal, we set its confidence score and its class probability to 1. For the NMS function, we use the default value of each parameter (see Appendix A.2 for details).

To measure the inference time of NMS for a specific number of bounding boxes (N), we set the confidence score to 1 for N proposals out of the 30,000 proposals. The remaining proposals (30,000 minus N) have their confidence score set to 0. Therefore, thanks to the confidence threshold, there will be N bounding boxes after NMS. In this study, the value of N ranges from 1 to 30,000. Also, it is important to know that models may have a threshold in terms of number of proposals such as YOLOv5 (25,000) and YOLOv8 (8,000). This factor may prevent a latency attack to succeed due to a potentially insufficient number of bounding boxes proposals passed to the NMS.

## E.2 Hardware: determining the success of an attack

We define a successful NMS attack as an attack capable to increase the processing time of the NMS above a defined threshold. In this paper, we set this threshold to 100 ms, 500 ms, 1000 ms, and 2000 ms. The value of each threshold serves as an indicator of the performance of the attack for a specific

use case. For instance, a maximum latency above 100 milliseconds is unsafe in the autonomous driving domain Lin et al. (2018). In a different context, a 100 ms delay will remain unnoticed by someone using an AI model for editing pictures on his phone or on his computer. However, a 2 seconds delay would be noticeable. Going back to Table 2, we observe a NMS attack will not work if the NMS processing is performed by a GPU instead of a CPU. Therefore, the evaluation shows that NMS latency attacks are harmless when NMS runs on GPU, which is becoming best practice.

Another factor impacting the success of the attack is the maximum number of proposal output by the AI model. As discussed in Appendix A.1 and shown in Table 2, YOLOv8 caps the number of proposals to 8,000. Hence, the attack can only generate sub-500 ms latency, considering it successful only for applications with latency threshold of 100-400 ms. For YOLOv5, which has a cap of 25,000 proposals, NMS attacks are capable to reach the 2 seconds threshold.

### E.3 Metrics

#### E.3.1 Metric: definition of intervals for number of bounding boxes (colors)

The definition of each interval is unique to this work. Our approach does not rely on empirical distributions, previous work, or some application-specific-criteria. The definition is purely based on arbitrary decision. To elaborate on the empirical distribution, this approach would have tied the definition of the intervals to the specific dataset used in our evaluation. Whereas, on the contrary, we wanted to have a metric dataset-agnostic. In our opinion, it was important to highlight at least the following scenarios: the attack does not work ('green'), the attack works ('red'), and the attack has some effect on the system while remaining unsuccessful ('yellow' and 'orange'). Using three intervals instead of four would be a valid alternative. At the end, the intention of our metric is to provides pointers for future research who wish to understand why an attack partially failed or succeed. Using a binary classification ('red' and 'green') would not provide such insights.

#### E.3.2 Additional metric: number of bounding boxes before and after the attack

Another way to provide more insights to the evaluation would be to compute the *boxes ratio under attack* (*brua*) (Equation 1) to assess the strength of the attack. This metric computes the ratio between the number of bounding boxes for the genuine image and the number of bounding boxes for the adversarial image. For instance, a genuine image contains 300 objects (bounding boxes) such as candies in a candy factory. Whereas, the adversarial version of this image contains 315 objects. Using our current evaluation, we may think the latency attack is working because there is a high number of bounding boxes. However, the high number of bounding boxes is due to the scene captured in the genuine image. Therefore, *brua* can be a handy metric to assess the attack strength in particular for an image or a dataset of images with large number of bounding boxes. Although, we believe *brua* will be better suited as a loss function to ensure the attack version of the image has more bounding boxes than the genuine version of the image.

$$brua = \frac{N_{bbox\_after\_attack}}{N_{bbox\_before\_attack} + \epsilon} \tag{1}$$

### E.4 Evaluation setup

Table 12 lists each setup used in our evaluation. As pointed out to us, in practice, latency can vary a lot depending on things like OS-level optimizations, driver versions, or background load. And, indeed, this study can be part of an extension for EVADE.

| GPU / CPU | Operating System | CUDA / Driver Version |
|---|---|---|
| NVIDIA A100-SXM4-80 GB / AMD EPYC | Ubuntu 20.04.6 | CUDA 12.2 |
| GRID A100D-10C / Intel Xeon | Ubuntu 20.04.6 | CUDA 12.2 |
| ORIGN / ARM Cortex | Ubuntu 20.04 | CUDA 11.4 |
| Apple M1 Max (CPU only) | macOS Sequoia 15.2 | N/A |

Table 12: Hardware and software configurations used in experiments

# F  Defenses

## F.1  Prior Art on defenses

In this paper, *PDM-Pure* is our active defense. *PDM-Pure* is an off-the-shelf adversarial purifier based on a pixel diffusion model (PDM). *PDM-Pure* aims to effectively eliminate adversarial patterns generated by latent diffusion models, thereby maintaining the integrity of images.

Our passive defense is *Maximum detection threshold*, a defense advertised in the *Overload* paper Chen et al. (2024). However, the impact of this defense on Overload has not been tested despite being already implemented as a parameter of the NMS component used by all YOLO models Jocher et al. (2023).

For our natural defense, we choose *JPEG*, a popular image compression algorithm, used in many use cases. For instance, an image may need to be compressed if the image needs to transits through a network with limited data bandwidth. This scenario can occur in embedded system such as within an autonomous driving car or in collective perception where a system shares images to an external party. During those uses cases, compression is necessary.

## F.2  Expanding the attack evaluation against defenses: model sizes and family

Table 13 extends the evaluation done in Section 4.7 by evaluating the performance of the defenses against attacks targeting models different than the original model (YOLOv8n) used in Table 7. These 2 models different from the original model on two aspects which are the model size (YOLOv8x) and the model family (YOLOv12n). Results-wise, Table 13 is consistent with Table 7 because each defense decreases drastically the number of bounding boxes. From an attack goal perspective, each defense prevents the attack to reach its goal which is a critical increase of inference time. However, preventing the attack's goal does not mean the defense prevented all goals of the attack. For instance in Table 13, even with a defense such as JPEG, an abnormally high number of bounding boxes remains. While not impact the inference time, this high number of bounding boxes still translate in poor detection performance because the model generates fake bounding boxes. This evaluation shows the importance of not choosing the defense from a single perspective (e.g., inference time).

Table 13: Model Comparison Under Various Defenses

| Defense | Model | Genuine | DS | O | PS | BPS |
|---------|-------|---------|-----|-----|-----|-----|
| None | v8n | 6 | 233 | 300 | 224 | 152 |
|  | v8x | 7 | 276 | 300 | 212 | 116 |
|  | v12n | 7 | 284 | 300 | 195 | 218 |
| PDM | v8n | 5 | 5 | 5 | 5 | 5 |
|  | v8x | 5 | 5 | 5 | 5 | 4 |
|  | v12n | 4 | 4 | 4 | 5 | 4 |
| JPEG | v8n | 6 | 8 | 55 | 5 | 6 |
|  | v8x | 6 | 8 | 9 | 7 | 7 |
|  | v12n | 6 | 64 | 21 | 4 | 4 |
| Max_Det(10) | v8n | 5 | 7 | 8 | 6 | 6 |
|  | v8x | 5 | 7 | 7 | 6 | 6 |
|  | v12n | 5 | 8 | 7 | 5 | 6 |

