# OpenReview forum: "Latency NMS Attacks: Is It Real Life or Is It Just Fantasy?"
_NeurIPS.cc/2025/Conference — NeurIPS 2025 poster_

### Official Review · Reviewer_TrTJ · 2025-06-25

**Clarity:** 3
**Significance:** 2
**Originality:** 3
**Rating:** 4
**Confidence:** 4

**Summary:**

The author proposes an evaluation framework named EVADE to assess the practicality of latency attacks, especially those targeting NMS in real-world scenarios, which is a novel direction. This framework comprises multiple components, providing a comprehensive combination of models, attacks, datasets, and more. It also employs different metrics to evaluate various aspects of the attacks. The extensive evaluation across different components reveals some interesting observations.

**Questions:**

1. While the authors refer to EVADE as a “framework,” I feel it is closer to a structured evaluation protocol or a testbed, as it lacks clear relationships between the components. In my view, a framework should be a more integrated and end-to-end system that not only provides a methodology for combining findings and reasoning about them but also offers an analytical structure to explain how different factors jointly influence the outcome.


2. The bar set for declaring an attack practical seems overly strict. In real-world systems, even an increase from 10ms to 50ms in the NMS step could break timing guarantees, especially in time-critical applications. Since object detection is typically just one stage in a tightly constrained processing pipeline, even small delays can compromise the overall system functionality. Evaluating latency variance, jitter, and system-level behavior under load would provide a more realistic assessment of an attack’s effectiveness.


3. On the platform side, the paper only lists the hardware type but doesn’t provide details about the actual deployment environment. In practice, latency can vary a lot depending on things like OS-level optimizations, driver versions, or background load. It would be helpful if the authors could clarify how sensitive their results are to these factors and whether the conclusions would still hold under different system configurations.


4. While the authors seem to treat lack of transferability as a sign that the attack isn’t effective, that mixes up two different things. Just because an attack doesn’t transfer well doesn’t mean it’s not dangerous, especially when the model is known or guessable.


5. What does “GRID A100D-10C” stand for in Table 2? Also, the Jetson AGX Orin has much stronger performance than typical edge devices, so a small latency increase on this platform may not reflect what would happen on more constrained hardware. It would be good to clarify whether the conclusions would still hold under truly resource-limited edge settings.


6. In the caption of Table 3, the authors mention using color “green (0–75), yellow (76–125), orange (126–225), red (226–300)” to divide the average number of bounding boxes. What’s the rationale behind choosing these specific thresholds? Are they based on empirical distributions, prior work, or some application-specific criteria?


7. While the results in Sections 4.2 and 4.3 show limited transferability across different model versions and sizes, it looks like the attacks are optimized separately for each model. It might be worth discussing whether known techniques for improving attack generalization could help with cross-model or cross-size robustness. Adding a short discussion on this could make the conclusions more broadly applicable.


8. In Section 4.7, the authors evaluate the defense methods using only one model, which makes it unclear whether the results generalize to other settings. Also, the evaluation focuses only on reducing the number of NMS outputs, without considering possible side effects of the defenses, such as degradation in performance on benign inputs. If a defense severely hurts normal performance, wouldn't that also count as a kind of attack success, even if latency remains unchanged?


9. In Section 4.8, the authors use a yes/no checklist over 6 or 7 criteria to conclude whether an attack is harmless. But this feels like an oversimplified evaluation. Just because an attack doesn't meet all the listed conditions doesn't mean it's completely ineffective or irrelevant. Some of the criteria don’t directly reflect actual impact, and the evaluation ignores cases where an attack could still be harmful in specific deployment settings. The overall reasoning seems too rigid to support such a broad conclusion.


10. While the paper claims to evaluate the real-world feasibility of NMS latency attacks, it mainly focuses on adversarial inputs. Other attack types like backdoors or bit-flips are dismissed as unrealistic, but in practice, these aren’t that uncommon, especially with social engineering or insider access. Also, just because an attacker has access to the model doesn’t mean they wouldn’t prefer a stealthy latency-based attack. The threat model feels a bit too narrow.

**Ethical Concerns:**

["NO or VERY MINOR ethics concerns only"]

**Final Justification:**

I appreciate the authors’ detailed responses and their willingness to revise the language in the paper to avoid overly strong terms such as “harmless” or “fantasy.” It's encouraging to see the authors recognize the need to frame the discussion more carefully, especially given the complexity of evaluating attack effectiveness across different deployment scenarios.

I have increased my score from 3 to 4 based on these clarifications and the authors’ commitment to revisions. Regarding Q2, I still feel that the paper would benefit from a clearer and more structured definition of what constitutes a “successful” attack under various system constraints. As suggested, including a table that explicitly outlines thresholds for latency increase under different deployment settings would make the paper more accessible and actionable for readers. I also support the idea of adding a brief discussion on system-level considerations, even if detailed evaluation is beyond the current scope. Overall, I appreciate the authors’ effort to engage constructively and improve the clarity and positioning of their work.

**Limitations:**

The discussion section does not clearly address the work's limitations. It mostly repeats the conclusion that the attacks are harmless, without reflecting on the scope or assumptions of the study. The suggestions for future work are vague and lack concrete methods or supporting evidence. A clearer discussion of what the current analysis does not cover would improve the paper.

**Paper Formatting Concerns:**

No major formatting issues noticed.

**Quality:**

2

**Strengths And Weaknesses:**

## Strengths:
1. This paper focuses on the real-world practicality of NMS-based latency attacks, which is a novel and valuable direction. It offers a new perspective for evaluating such attacks and lays a solid foundation for future research in this area.

2. The proposed EVADE framework consists of multiple components and configurations, serving as a flexible and effective testbed for adversarial image-based latency attacks.

3. The paper presents a well-structured set of research questions, which are carefully designed to cover diverse evaluation dimensions. This provides a clear and thoughtful basis for analyzing the effectiveness of latency attacks.

4. The experimental evaluation is extensive and well-aligned with the research questions. Each component is evaluated through detailed experiments, which demonstrate substantial effort and contribute to a comprehensive analysis of the proposed framework.

## Weaknesses:
1. The author calls EVADE a framework, but it feels more like a collection of inputs and questions. There’s no clear internal structure connecting the parts, which makes it look more like a testbed than a true framework.

2. The paper only focuses on adversarial image attacks at the input level, which limits the scope of the work.

3. The paper sets a very high bar for what counts as a “successful latency attack,” and some of the main conclusions are stated quite definitively, even though the evidence doesn’t always seem strong enough to fully support them.

4. The analysis mainly stays at the model level, without a broader system-level view, which is quite important when considering real-world practicality.

5. Most importantly, the authors didn't prove that NMS latency attacks are harmless. Instead, they show that some existing attack methods fail under specific experimental settings, and from that, draw a broad conclusion that the overall attack idea is ineffective. However, the failure of current attacks does not mean the general approach is invalid. Similarly, lack of transferability does not imply real systems are safe, and partial defense success does not prove the attack is harmless. Overall, the conclusion feels overstated compared to the evidence provided.

---

> ### Author Rebuttal · Authors · 2025-07-30
>
> Dear reviewer,
>
> Thank you for your time in providing constructive feedback on our submission. We appreciate your insights and would like to address your concerns and your questions.
>
> ## Q1
>
> The reason for considering EVADE as a "framework" was that **our methodology integrates the evaluation from multiple components**. As detailed in Section 3 and seen in Figure 2, EVADE comprises distinct, orthogonal components (Hardware, Model Family, Quantization, etc.). The results from evaluating each of these components are not presented in isolation. Instead, they are **systematically aggregated** into a final score and a conclusive, practical assessment, as demonstrated in our summary as shown in Table 9.
>
> Also, the framework is **extensible** -- it can incorporate new metrics (e.g., image similarity scores), new evaluation modules (e.g., at the Operating System level), and new domains like Natural Language Processing (NLP) and Audio Speech Recognition (ASR).
>
> In the revised manuscript, we will add a paragraph in Section 3 to explicitly define our use of the term "framework," emphasizing both the integration of findings (as shown in Table 9) and the extensible nature of the components to better align with the reviewer's valuable perspective. **Alternatively, we are open to replacing "framework" with "evaluation protocol" if that better reflects the intended meaning**.
>
> ## Q2
>
> Good point, it is true the attack is effective when latency increases from 10 ms to 50 ms. Our framework is designed to be flexible; a user with stricter latency requirements (e.g., 50ms) can use our findings to determine how many proposals are needed to exceed the targeted threshold for a successful attack and evaluate the attack's risk accordingly. For example, Table 2 reflects this observation. if the NMS inference time increases beyond 100 ms (or even 10 ms or 50 ms), the attack is effective.
>
> We agree that evaluating the attack from a system-level perspective is important. However, most latency attack papers claim effectiveness without considering system-specific factors. To align with their evaluation scope, we have not included a system-level evaluation at this stage.
>
> ## Q3
>
> We appreciate the reviewer's insightful comment regarding the detailed characterization of the deployment environment and its potential impact on latency measurements. Below is the list of OS and driver versions used in our paper:
>
> 	- Setup #1 (NVIDIA A100-SXM4-80 GB (GPU) & AMD EPYC (CPU))
> 		Ubuntu 20.04.6 (OS)
> 		CUDA version 12.2 (driver).
>
> 	- Setup #2 (GRID A100D-10C (GPU) & Intel Xeon (CPU)),
> 		Ubuntu 20.04.6 (OS)
> 		CUDA version 12.2 (driver).
>
> 	- Setup #3 (ORIGN (GPU) and ARM Cortex (CPU))
> 		Ubuntu 20.04 (OS)
> 		CUDA 11.4 (driver).
>
> 	- Setup #4 (Apple M1 Max CPU),
> 		Sequoia 15.2 (OS).
>
> Also, we acknowledge that factors such as OS-level optimizations, specific driver versions, and background system load can indeed influence the observed latency. However, delving into the exhaustive analysis of every possible permutation of operating systems and drivers is:
> - **beyond the scope of this paper**,
> - **not a mandatory requirement (according to prior art) for the success of an attack**
> - **beyond our available resources allocated to this initial study**
>
>
> Therefore, we plan to investigate those aspects in future iterations of the EVADE framework.
>
> ## Q4
>
> Yes, we agree with the reviewer's observation and we have the same observation as seen in Tables 3 and 4. The attack is ineffective (i.e., does not transfer) when targeting a model different from the one used to optimize the adversarial input. As expected from a white-box attack, the attack is only successful when targeting the same model.
>
> We will clarify that while low transferability limits the efficacy and generalizability of latency attacks, it does not diminish the threat posed by latency attacks against known models.
>
> ## Q5
>
> "GRID A100D-10C" refers to NVIDIA vGPU (Virtual GPU) A100D-10C. This configuration allocates a virtual A100 GPU instance with 10 GB of GPU memory and is commonly used in cloud-based deployments for efficient resource sharing. We will clarify this abbreviation in the updated version of Table 2 and the surrounding text.
>
> Regarding Jetson AGX Orin and resource-limited edge devices, we agree that the Jetson AGX Orin is a relatively high-performance edge device with growing adoption in more demanding edge AI applications. While we did not test on ultra-low-power devices, **our results on the Orin suggest that these specific attacks are generally impractical on such platforms.**
>
> For more constrained devices like the Jetson NX, Table 3 of the Overload paper [1] shows that inference time can peak at 230 ms under attack. This implies that, based on our Table 2, the attack is effective if the application requires inference times below 230 ms, but harmless if the threshold is higher (e.g., 500, 1000, or 2000 ms). We will add a discussion to the manuscript to highlight this point.
>
> [1] Chen, Erh-Chung, et al. "Overload: Latency attacks on object detection for edge devices." CVPR. 2024.
>
> ## Q6
>
> For now, the interval are purely informative. The key takeaway is:
>  - the attack is effective if the attack is tagged as `red`
>  - the attack is ineffective if the attack is tagged as `green`, `yellow`, or `orange`
>
>
> Colors `yellow` and `orange` highlight images where the attack had some success in transferability. This information provides pointers for future research who wish to understand why. Such pointers would not exist if we had chosen a binary classification (`red` and `green`).
>
>
> ## Q7
>
> Yes all the attacks are white-box attacks and thus, they are optimized for a single model.
>
> We agree with the reviewer and will add a discussion section highlighting the need to explore transferability boosting techniques such as QAA [2] and MI [3] to see if those attacks are more successful.
>
> [2] Yang, Yulong, et al. "Quantization aware attack: Enhancing transferable adversarial attacks by model quantization." IEEE Transactions on Information Forensics and Security 19 (2024).
> [3] Yinpeng Dong, et al. Boosting adversarial attacks with momentum. CVPR, 2018
>
> ## Q8
>
> We tested various model sizes and versions and consistently observed the same outcome.
>
> ## Table: Model Comparison Under Various Defenses
>
> |Defense|Model|Genuine|Daedalus|Overload|Phantom Sponge | Beyond Phantom Sponges |
> |-|-|-|--|-|-|-|
> |None|v8n|6|233|300|224|152|
> ||v8x|7|276|300|212|116|
> ||v12n|7|284|300|195|218|
> |PDM|v8n|5|5|5|5|5|
> ||v8x|5|5|5|5|4|
> ||v12n|4|4|4| 5|4|
> |JPEG|v8n|6|8|55|5|6|
> ||v8x|6|8|9|7|7|
> ||v12n|6|64|21|4|4|
> |Max_Det(10)|v8n|5|7|8|6|6|
> ||v8x|5|7|7|6|6|
> ||v12n|5|8|7|5|6|
>
>
> These defenses consistently generalize across different model types and sizes, significantly reducing the number of bounding boxes.
>
> We acknowledge that some defenses may degrade system performance. However, this is a limitation of the defense itself, not the attack. Our goal was to demonstrate that there are multiple ways to mitigate the attack’s objective—namely, increasing inference time.
>
> We also want to point out that the strong reduction in mAP caused by poorly designed defenses has never been adequately addressed in previous latency attack papers. Notably, after our submission, a team published a defense specifically targeting latency attacks [4]. Their approach successfully removes adversarial bounding boxes while preserving mAP performance. This shows that effective defenses can mitigate latency attacks without compromising detection accuracy.
>
> [4] Wang, Tianyi, et al. "Can't Slow Me Down: Learning Robust and Hardware-Adaptive Object Detectors against Latency Attacks for Edge Devices." Proceedings of the Computer Vision and Pattern Recognition Conference. 2025.
>
>
> ## Q9
>
> We agree that the attack could work under very specific conditions (e.g., non-quantized, no defense, the attacker knows the model). This makes the attack relatively niche, as it relies on strong assumptions about the system.
>
> Our goal with EVADE was to establish practical benchmarks based on realistic deployment scenarios. The "harmless" conclusion was drawn in the context of exposing the broader challenges and to disconnect between theoretically potent attacks and their practical impact in real-world systems.
>
> We will clarify this motivation in final manuscript, emphasizing that the checklist serves as a pragmatic framework to assess the general practicality of attacks in the common scenarios we investigated, thereby encouraging future research to account for these real-world constraints from the outset.
>
> ## Q10
>
> About the attacker model, we intended to include the latency backdoor attack and the latency bit-flip attack. However, the papers did not provide their code. As a matter of fact, we contacted the authors without success. We could reproduce their work but then people could challenge the fidelity of our reproduction of the attack.
>
> We agree that this remains a promising direction for future exploration.

---

> > ### Comment · Reviewer_TrTJ · 2025-08-04
> >
> > Thank you to the authors for the detailed responses. I’m generally fine with the replies to Q1, Q3, Q4, Q7, and Q10.
> >
> > For Q2, I feel the paper lacks a clear definition of what constitutes a successful attack under different deployment scenarios. For instance, in the response to Q6, the authors suggest that exceeding 230 ms can be considered a success, but the overall framing of the paper implies that only much larger latency increases qualify. There is also no explicit discussion of system-level behavior or timing constraints.
> >
> > For Q5, I remain unconvinced by the claim that the Jetson AGX Orin is representative of more lightweight edge devices. Concluding that the attack is impractical on constrained hardware based solely on Orin results seems insufficient.
> >
> > Regarding Q6, I now understand that the bounding box thresholds are meant to be informative, but they still appear arbitrarily chosen rather than grounded in empirical data or application requirements.
> >
> > For Q8, I agree that poor defense design shouldn’t be blamed on the attack. That said, if a defense severely degrades performance on benign inputs, it may still be considered part of the attack’s overall impact from a systems perspective.
> >
> > For Q9, I’d suggest softening the checklist-based conclusion. Some of the criteria may not directly reflect actual harm, and a strict yes/no framework could miss meaningful edge cases. Clarifying that the checklist is a heuristic rather than a definitive test would make the conclusions more balanced.
> >
> > My main concern remains the strong categorical claims like "harmless" or "fantasy," which feel premature without more clearly defined success criteria across diverse deployment settings. If this part were revised or clarified, I’d be happy to raise my score.

---

> > > ### Author Response · Authors · 2025-08-04
> > > **We will soften the tone and focus on highlighting the challenges.**
> > >
> > > Thank you for your thoughtful feedback. We appreciate your point regarding the categorical language used in our paper. Our intention was to emphasize the practical challenges and limitations of NMS latency attacks in real-world scenarios, not to dismiss their potential impact under specific conditions.
> > >
> > > In response to your concern, we will revise the wording to soften terms like “harmless” and “fantasy”. We agree it is important to ensure that we do not diminish the effectiveness of such attacks where they are applicable. Also, we will focus on highlighting the challenges these attacks face across diverse deployment settings, which we believe is a valuable direction for future research.
> > >
> > > >For Q2, I feel the paper lacks a clear definition of what constitutes a successful attack under different deployment scenarios. For instance, in the response to Q6, the authors suggest that exceeding 230 ms can be considered a success, but the overall framing of the paper implies that only much larger latency increases qualify. There is also no explicit discussion of system-level behavior or timing constraints.
> > >
> > > Sorry for the confusion, as stated in Table 2, attacks targeting GPU hardware are successful only if the latency requirements are small (e.g., 230 ms and under). However, those attacks, targeting GPU, are not successful for higher latency requirements (e.g 500 ms above). We can rewrite the section the paper about "only much larger latency increases qualify" to remove any confusion. Regarding the impact of the attack on system-level behavior and timing constraints, we can add a discussion section explaining further evaluation needs to explore the impact of latency attacks beyond the scope of the model by doing system-level evaluation. For instance, system-level evaluation would help to answer the following questions:
> > >
> > > - how an unsuccessful attack can delay downstream tasks (e.g., Object Tracking [1])?
> > > - how OS and driver configurations can affect the inference time?
> > > - how background tasks could affect the inference time of the Object detection model?
> > >
> > > [1] Muller, Raymond, et al. "Investigating physical latency attacks against camera-based perception." 2025 IEEE Symposium on Security and Privacy (SP). IEEE, 2025.
> > >
> > > >For Q5, I remain unconvinced by the claim that the Jetson AGX Orin is representative of more lightweight edge devices. Concluding that the attack is impractical on constrained hardware based solely on Orin results seems insufficient.
> > >
> > > Fair enough, we can say our evaluation was done for devices ranging from powerful edge devices (e.g., ORIN) to servers. However, there is still a need to evaluate smaller hardware running on a more resources constrained IoT device or phone such as Jetson Nano, Qualcomm RB3.
> > >
> > > >Regarding Q6, I now understand that the bounding box thresholds are meant to be informative, but they still appear arbitrarily chosen rather than grounded in empirical data or application requirements.
> > >
> > > Yes, they are. We can highlight this limitation and say room to improve such metric with a more grounded approach such as empirical data or application requirements
> > >
> > > >For Q8, I agree that poor defense design shouldn’t be blamed on the attack. That said, if a defense severely degrades performance on benign inputs, it may still be considered part of the attack’s overall impact from a systems perspective.
> > >
> > > Agreed, we can include this perspective into the paper. Despite mitigating the increase of latency, the inclusion of, certain, defenses can lead to an overall reduction of mAP. For instance, as seen in [2], on genuine data, the system with the defense has a worse mAP score than the system without the defense. Under adversarial attacks, fake bounding boxes may remain, despite the presence of a defense mechanism, and result in a decrease of mAP.
> > >
> > > [2] Wang, Tianyi, et al. "Can't Slow Me Down: Learning Robust and Hardware-Adaptive Object Detectors against Latency Attacks for Edge Devices." Proceedings of the Computer Vision and Pattern Recognition Conference. 2025.
> > >
> > > >For Q9, I’d suggest softening the checklist-based conclusion. Some of the criteria may not directly reflect actual harm, and a strict yes/no framework could miss meaningful edge cases. Clarifying that the checklist is a heuristic rather than a definitive test would make the conclusions more balanced.
> > >
> > > Agreed, Table 9 is an example on how to make use of our evaluation. We could add an additional raw to illustrate the an alternative scenario. Assume someone wants to focus on the following modules : Hardware (Threshold<100ms), Quantization, and Format. Then, Overload will have a score of 3/3 and the other attacks will have a score of 2/3. In such scenario, the attacks are worth considering.

---

### Official Review · Reviewer_3QsT · 2025-07-01

**Clarity:** 3
**Significance:** 3
**Originality:** 3
**Rating:** 5
**Confidence:** 3

**Summary:**

This paper challenges the practicability of the NMS attacks, which aims to increase the latency of object detector by increase the number of object proposals. To challenges the effectiveness of these existing attacks, this work proposed a new evaluation benchmark that studies the effectiveness of the existing attacks when (1) hardware changes, (2) architecture changes, (3) model size changes, (4) model format changes, (5) model is quantized, (6) input image has domain shift and (7) model has defense mechanism. The entire paper focus on the experimental discoveries. It concludes that the NMS attack remains effective when the model format and quantization changes and the NMS attack is no longer effective when the remaining factor changes.

**Questions:**

Please refer to the weakness section.

**Ethical Concerns:**

["NO or VERY MINOR ethics concerns only"]

**Final Justification:**

Thanks the author for providing the rebuttal. After reading the rebuttal, I remain the score as I still think the paper is interesting and this is one of the pioneering works to provide this study (from my best understanding). While there are many attacks proposed in the literature, whether these attacks are practical in the real-world scenario remains unknown, and this is where this paper contributes.

Yet, as mentioned by other reviewers and also the initial comment, this contribution of this work is the testbench and evaluation protocol itself. Again it remains questionable whether this is beyond Neurips acceptance.

Overall, I still think this is a good work

**Limitations:**

Yes, the author has addressed the limitations of this work and proposed serveral directions to expand the proposed dataset.

**Quality:**

3

**Strengths And Weaknesses:**

Strength
•	The paper is well organized and easy to follow and read.
•	The evaluation benchmark can be helpful for the larger community for testing whether the new developed NMS attack is generalizable.
•	This paper seems to be the pioneering in studying the effectiveness of existing NMS attack.
•	The experiments investigate the effectiveness of the attack in multiple factors, which are well designed.

Weakness
•	This paper is a discovery type of work, where only observation is presented but there is not further explanation about why the attack fails or how to make the attack more effective.  For example, why switching across different hardware will decrease the effectiveness of the attack.
•	The only contribution is the evaluation benchmark itself (perhaps also the conclusion), but it remains debatable whether this is sufficient for Neurips acceptance
•	Will the author release the evaluation benchmark for reproducibility? For the “Hardware” component, how does other researcher in the community reproduce the result without having the hardware? This is important as the benchmark is the core and only contribution of this work.
•	In section 3.3, the author categorizes the number of bbox into 4 different ranges. However, it is, by nature, that some of the images in the evaluation benchmark has more bounding box (i.e. objects) than other images, without being attacked. How does the metric account for this? Doesn’t it make more sense to produce a normalize score, where the normalized score is (# bbox after attack) / (# of bbox before attack) ?

---

> ### Author Rebuttal · Authors · 2025-07-30
>
> Dear reviewer,
>
> Thank you for taking the time to provide constructive feedback on our submission. We appreciate your insights and would like to address the concerns and questions you raised in your review.
>
> # Weakness
>
> > This paper is a discovery type of work, where only observation is presented but there is not further explanation about why the attack fails or how to make the attack more effective. For example, why switching across different hardware will decrease the effectiveness of the attack.
>
> Our primary objective with EVADE is to highlight the broader challenges and disconnects between theoretically strong attacks and their practical effectiveness in real-world systems. We are particularly interested in exploring why these challenges arise and how they can be addressed in future research.
>
> > Will the author release the evaluation benchmark for reproducibility? For the “Hardware” component, how does other researcher in the community reproduce the result without having the hardware? This is important as the benchmark is the core and only contribution of this work.
>
> Yes, we plan to release the benchmark and dataset upon publication of the paper, and we will include a link in the final manuscript.
> Regarding hardware requirements, EVADE can be applied to any hardware researchers wish to evaluate. The hardware listed in our paper serves as a representative sample to demonstrate the challenges of current attacks.
>
> > In section 3.3, the author categorizes the number of bbox into 4 different ranges. However, it is, by nature, that some of the images in the evaluation benchmark has more bounding box (i.e. objects) than other images, without being attacked. How does the metric account for this? Doesn’t it make more sense to produce a normalize score, where the normalized score is (# bbox after attack) / (# of bbox before attack) ?
>
> The attacks we evaluated aim to generate a high number of additional bounding boxes, far exceeding the natural count in any typical image. Therefore, our ranges were set to assess how many additional, fake bounding boxes the attack could force the system to process.
> However, we agree that a normalized score, such as $\frac{N\\_bbox\\_after\\_attack}{N\\_bbox\\_before\\_attack}$, offers a valuable perspective, particularly for understanding the relative impact on individual images, especially those with a naturally higher baseline. Actually, we can also add a variable $\epsilon$ to the denominator to cover the case where a genuine image may have no bounding box (to prevent a division by 0) such as $\frac{N\\_bbox\\_after\\_attack}{N\\_bbox\\_before\\_attack\ +\ \\epsilon}$
> We will clarify this observation and incorporate a discussion on normalized score to address this issue.

---

### Official Review · Reviewer_eEBD · 2025-07-03

**Clarity:** 2
**Significance:** 1
**Originality:** 3
**Rating:** 4
**Confidence:** 3

**Summary:**

### Summary
The paper investigates the transferability and robustness of NMS latency attacks on object detection models. The authors evaluate the effectiveness of these attacks across different model formats, quantization levels , and versions of the same model. They also examine how defenses impact the success rate of these attacks.

**Questions:**

The main conclusion—that NMS latency attacks are generally ineffective—might be seen as discouraging for those looking for practical adversarial strategies. However, this is balanced by the identification of new research avenues. Somewhat, these methods provide insight on the robustness of NMS mechanism. Do you have more insightful suggestions?

**Ethical Concerns:**

["NO or VERY MINOR ethics concerns only"]

**Final Justification:**

The author's rebuttal solved my concerns; I raise my rating to acceptance.

**Limitations:**

1. While the paper suggests exploring NMS-free models, it does not delve into developing specific attacks for these models. Creating and evaluating new attack vectors tailored for NMS-free architectures would be a valuable extension.
2. More suitable for the position paper track.

**Paper Formatting Concerns:**

Nothing.

**Quality:**

2

**Strengths And Weaknesses:**

### Strengths

1. They perform comprehensive experiments. The study uses an evaluation framework -  EVADE to systematically assess various aspects of NMS latency attacks, including transferability, quantization effects, and defense mechanisms.
2. The experiments involve real-world datasets such as COCO 2017 and consider practical factors like model conversion and optimization.
3. The paper provides valuable insights into the limitations of current NMS latency attacks and suggests directions for future research, particularly in targeting NMS-free models or analyzing post-object detection processes.

### Weaknesses
1. The paper does not provide a comprehensive study of quantization techniques, which could limit its applicability in some contexts.
2. While the paper identifies the need for more effective attacks, it does not propose concrete solutions or improvements.

---

> ### Author Rebuttal · Authors · 2025-07-30
>
> Dear reviewer,
>
> Thank you for taking the time to provide constructive feedback on our submission. We appreciate your insights and would like to address the concerns and questions you raised in your review.
>
> ## Question:
>
> > The main conclusion—that NMS latency attacks are generally ineffective—might be seen as discouraging for those looking for practical adversarial strategies. However, this is balanced by the identification of new research avenues. Somewhat, these methods provide insight on the robustness of NMS mechanism. Do you have more insightful suggestions?
>
> **TLDR**:
> - Our conclusion aims to provide **a different point of view and new research directions**
> - We have provided 2 insights to tackle existing limitations:
>   - **`The first insight` provides directions to boost latency attacks transferability**
>   - **`The second insight` provides a new attack goal for latency attacks**
>
>
> **Detailed answer**
>
> Certainly, we can provide some suggestions.
>
> The major limitation of all the NMS attacks is that they are white-box attacks. As a result, these attacks are optimized to work for a single model, as demonstrated in our transferability results.
>
> On the contrary, we see this paper as an opportunity to highlight the practical limitations encountered when deploying these attacks against object detection models in production. These limitations can serve as new research directions for developping more effective latency attacks. Such challenges have been largely overlooked in previous work. One possible reason is that prior research has focused on maximizing attack strength, often assuming no practical constraints on deployment. After several years and numerous papers on this topic,our goal with this paper is toraise awareness in the research community about these real-world limitations, which have not been previously addressed.
>
>
> That said, there are several ways to tackle these challenges. We share 2 insights for future research.
>
> 1. **Improving Attack Transferability**:
> One solution could be to develop a black-box version of the NMS attack, such as an optimization-free attack [1]. his would avoid tailoring the attack to a specific model and instead focus on identifying adversarial patterns that are model-agnostic, thereby improving transferability. Another approach could involve combining existing NMS attacks with transferability-boosting techniques such as `Quantization Aware Attack (QAA)` [2] and `Momentum-based iterative (MI)` [3]
>
> 2. **Generating High-Confidence Fake Boxes**:
> Most fake bounding boxes currently have lower confidence scores than genuine ones. As a result, defenses that limit the number of displayed boxes tend to retain genuine boxes, since they have higher confidence. If an attack could generate fake boxes with higher confidence than the genuine ones, the defense might mistakenly discard the genuine boxes.
>
> We would be happy to include these insights in a discussion section.
>
> **References**
> [1] Liu, Hangcheng, et al. "Optimization-Free Patch Attack on Stereo Depth Estimation." arXiv preprint arXiv:2506.17632 (2025).
> [2] Yang, Yulong, et al. "Quantization aware attack: Enhancing transferable adversarial attacks by model quantization." IEEE Transactions on Information Forensics and Security, 2024.
> [3] Yinpeng Dong, et al. Boosting adversarial attacks with momentum. CVPR, 2018.
>
> ## Weakness and limitations:
>
> > The paper does not provide a comprehensive study of quantization techniques, which could limit its applicability in some contexts.
>
> Our primary objective in including quantization was to highlight it as a realistic and practical factor that significantly impacts the effectiveness of current NMS latency attacks in real-world deployments. We showed that even basic quantization strategies can reduce attack effectiveness. We agree that a comprehensive analysis of various quantization methods (e.g., different bit-widths, post-training vs. quantization-aware training, mixed-precision) is a valuable direction for future research.
>
> > While the paper identifies the need for more effective attacks, it does not propose concrete solutions or improvements.
>
> The main objective of this work is to introduce EVADE, a framework for evaluating the effectiveness of current latency attacks. The limitations we raise point to new challenges that future work can address. Additionally, we have provided insights into how these challenges might be tackled (see the reponse provided in the question section above)
>
> > While the paper suggests exploring NMS-free models, it does not delve into developing specific attacks for these models. Creating and evaluating new attack vectors tailored for NMS-free architectures would be a valuable extension.
>
> Our paper has two goals:
> - questions the effectiveness of existing latency attacks
> - proposes a new way to evaluate latency attacks
>
> Designing a latency NMS-free attack would require a dedicated paper, as it would involve
> - introducing the design of the attack
> - presenting the evaluation methodology (EVADE)
> - Analyzing the attack's performance.
>
> Due to space constraints, we could not include such an attack in this paper. However, we do plan to pursue this direction in future work.
>
> As highlighted in Section 5.2, valuable extensions to our work could be:
>  - to expand EVADE to other types of attacks such as misclassification attacks and removal attacks
>  - to expand EVADE to other types of modalities such as text and audio
>
> > More suitable for the position paper track
>
> We were indeed debating between the evaluation track and the position paper track. Either option works for us!

---

> > ### Author Response · Authors · 2025-08-06
> > **Rebuttal discussion**
> >
> > Dear Reviewer,
> >
> > We hope our rebuttal has addressed your questions and concerns effectively. We would be grateful for any feedback or response you could share, as it would help us better understand your perspective and further improve our work.
> >
> > Thank you again for your time and consideration.
> >
> > Best regards

---

> > > ### Comment · Area_Chair_roM3 · 2025-08-07
> > >
> > > Dear Reviewer,
> > >
> > > The authors have submitted a rebuttal. Can you please provide your opinion after reading the rebuttal?
> > >
> > > Thanks,
> > > AC

---

### Note · Authors · 2025-08-11

We thank the reviewers and AC for their thoughtful feedback and constructive suggestions. Our primary goal with this work is to bridge the gap between theoretically strong latency attacks and their practical implications in real-world deployments. While prior research has largely focused on maximizing attack strength under idealized conditions, our study highlights critical limitations—such as hardware diversity, quantization, and defense mechanisms—that significantly affect attack feasibility.

# Key clarifications and contributions

##
**EVADE as a framework**: EVADE integrates multiple orthogonal components (hardware, model family, quantization) into a unified **evaluation protocol**, producing a practical assessment rather than isolated metrics. It is extensible to new domains (e.g., NLP, ASR) and evaluation modules.

**Insights for tackling existing limitations**: We identify two promising ways:
* improving attack transferability via black-box or optimization-free approaches and transferability-boosting techniques (e.g., QAA, MI)
* generating high-confidence fake boxes to bypass current defenses.

**Scope and limitations**: Our findings do not dismiss the potential impact of latency attacks under specific conditions; rather, they emphasize the practical challenges these attacks face in diverse deployment scenarios. We will soften categorical language (e.g., “harmless”) and clarify that our checklist serves as a heuristic, not a definitive test.

**Benchmark release**: We confirm that the benchmark and dataset will be publicly released upon publication to ensure reproducibility.

**Future direction**: We plan to extend EVADE to other attack types (e.g., misclassification, removal) and modalities (text, audio), explore system-level evaluations, and investigate attacks on more resource-constrained devices.

In summary, this work provides the first systematic evaluation of latency attacks under realistic constraints, offering a foundation for future research to design more practical and robust adversarial strategies. We believe these contributions align with the community’s need for rigorous, deployment-aware security assessments.

---

### Decision · Program_Chairs · 2025-09-17

**Decision:**

Accept (poster)

**Comment:**

All reviewers suggest accepting the paper, so will be accepted. Please address the remaining concerns in the camera ready version.